# Salt Distribution and Potato Response to Irrigation Regimes under Varying Mulching Materials

**DOI:** 10.3390/plants9060701

**Published:** 2020-05-31

**Authors:** Mohamed Hassan Abd El-Wahed, Abdulrasoul Mosa Al-Omran, Mahmoud Mohamed Hegazi, Mahmoud Mohamed Ali, Yahia Abdelaty Mohamed Ibrahim, Ayman EL Sabagh

**Affiliations:** 1Arid Land Agriculture Department, Faculty of Meteorology, Environment & Arid Land Agriculture, King Abdulaziz University, Jeddah 80208, Saudi Arabia; abdelwahed_m64@yahoo.com; 2Soil Science Department, College of Food and Agricultural Sciences, King Saud University, P.O. Box 2460, Riyadh 11451, Saudi Arabia; rasoul@ksu.edu.sa; 3Agricultural Engineering Department, Faculty of Agriculture, Ain Shams University, Cairo 11566, Egypt; mahmoudhegazi47@gmail.com; 4Agricultural Engineering Department, Faculty of Agriculture, Fayoum University, Faiyum 63514, Egypt; mma12@fayoum.edu.eg (M.M.A.); engyahia2022@yahoo.com (Y.A.M.I.); 5Agronomy Department, Faculty of Agriculture, Kafrelsheikh University, Kafr El-Sheikh 33516, Egypt; 6Department of Field Crops, Faculty of Agriculture, Siirt University, 56100 Siirt, Turkey

**Keywords:** deficit irrigation, mulching, potato yield, salt distribution

## Abstract

Water scarcity and frequent drought spells are becoming critical challenges to sustainable agricultural development, especially in arid and semiarid regions. Thus, this work aims to investigate the effect of deficit irrigation and varying mulching materials on soil moisture content, salt distribution, and potato yield. The experiment consisted of three irrigation regimes (I_100%_, I_80%_, and I_60%_) of crop evapotranspiration (ETc), designated as I_100%_, I_80%_, and I_60%_ of ETc, and five mulching treatments viz. (i) without mulch (WM), (ii) poultry manure mulch (PMM), (iii) rice straw mulch (RSM), (iv) white plastic mulch (WPM), and (v) black plastic mulch (BPM), which were continued for two consecutive growing seasons. The results showed that soil salinity was affected by mulching and irrigation levels as the salt content increased from the initial soil salinity. Moreover, I_60%_ recorded the highest salt accumulation in the soil profile for WM treatment compared to the rest of the irrigation and mulching treatments. It was also revealed that PMM remained unmatched by significantly producing the highest potato yield compared to other mulching materials. However, the average potato yield decreased by 13.83% and 29.16% in the 2016 season for I_80%_ and I_60%_ and by 12.95% and 30.91% in the 2017 season, respectively, in comparison to full irrigation (I100%). So, when sufficient irrigation water is available, full irrigation (I_100%_) and PMM treatment are recommended to achieve the maximum potato tuber yield, which has a minimum impact on increasing salinity. However, when the discharge is insufficient, deficit irrigation (I_80%_) and PMM treatment are recommended to conserve 20% of the irrigation water applied with a minimum reduction in tuber yield and a slight increase in soil salinity.

## 1. Introduction

In Egypt, more than 80% of water resources are used for irrigating crops of immense economic significance [1], which ensure the livelihood and food security of the populace. However, climate change characterized by rising temperatures and shifting of rainfall patterns and distribution has led to freshwater shortages, especially in arid and semiarid regions [2,3]. Hence, the water shortage has emerged as one of the vital problems of and the most serious challenge to the sustainable development of agriculture [4]. As a result, eco-friendly, biologically viable, and economically doable agricultural practices must be developed to conserve soil moisture. Conservation of irrigation water and increases in crop productivity are two interrelated challenges in Egypt [5].

Globally, potato (*Solanum tuberosum* L.) is the third most cultivated food crop after rice and wheat [6], and it constitutes the staple food of many nations. In Egypt, it is the second most cultivated vegetable crop after tomato. Recently, Egypt has been ranked at the top among potato exporters in the world [7]. Potato is very sensitive to soil–water conditions, as water stress (drought and flooding) leads to a serious reduction in the yield and quality of potato tubers [8]. Therefore, irrigation management becomes pivotal for potato production, as per the varietal potential.

To maintain crop productivity under drought in arid regions, deficit irrigation (DI) has been widely studied for many food and cash crops [9]. It aims to increase crop water productivity (CWP; the relationship between the crop produced and the amount of irrigation water involved in crop production) due to increased application efficiency since most of the applied water remains in the root zone [10]. In addition, DI tends to produce equivalent potato yields to full irrigation (FI), and has been shown to improved CWP by 60% by conserving irrigation water up to ~30% [11]. The effect of DI on the physiological characteristics of potato has been studied and the results clearly indicate that the relative chlorophyll contents of potato plants remain at par to FI [12]. Fresh yields and the CWP of potatoes are significantly affected by DI in comparison to the effect of FI [13]. However, a research gap exists regarding potato response to DI under the agro-ecological conditions of Egypt, which necessitates conducting studies to obtain fresh field evidence.

Among water conservation techniques, mulching (covering of the soil surface with biotic and/or synthetic materials) has been reported to reduce water losses through evaporation, thereby enhancing plant growth and development. Different types of mulching materials, including (i) organic materials (e.g., crop residues, straws, grasses, poultry manure, and farmyard manure) and (ii) inorganic materials (e.g., polyethylene sheets and gravels), have been put into use with varying and contrasting results on water conservation and crops yield. Mulching improves the water status in soil [14] by inhibiting evaporation through shielding the direct exposure of sunlight to the soil’s upper horizons [15], leading to increased crop yield [16]. For different mulch treatments, some researchers have recorded the highest soil salinity value when no mulching is applied in comparison with other mulching treatments, based on reduced evaporation from the soil surface as well as regulated soil water and salt movement [17,18]. Soil salinity has been reported to multiply with increasing deficit irrigation [17,19].

It has been hypothesized that deficit irrigation, in conjunction with mulching, might produce comparable potato yields as that of full irrigation. Therefore, this work objectively evaluates the effect of deficit trickle irrigation and different mulching materials on yield component, potato tuber yield, crop water productivity, and salt distribution.

## 2. Materials and Methods

### 2.1. Experimental Location

Field experiments were carried out during the 2016 and 2017 seasons at the experimental station of the Faculty of Agriculture, Fayoum University (FU), Egypt (longitude: 30° 85′ E, latitude: 29° 30′ N, altitude: 25 m, and fetch of short vegetation around the evaporation pan: 1000 m). Soil analyses were conducted in the soil testing laboratory of the Faculty of Agriculture, FU, at the beginning of experiments. Table 1 and Table 2 show some physical and chemical properties of the soil at the station.

### 2.2. Experimental Design and Treatments

Three replications of each treatment arranged in a randomized complete block were conducted while the potato crop was irrigated using a trickle irrigation system. The experimental field was divided into three main irrigation treatment plots, where each plot had one valve (on and off). Furthermore, each main plot had five mulching treatment subplots with three laterals, where a single lateral existed per potato row. Each lateral had 16 mm diameter, 16 m length, and emitters spaced 30 cm apart. Each emitter flow rate was 2.l Lh^−1^ at a pressure of 1 bar, and the laterals were spaced at 70 cm.

The irrigation treatment plots were isolated with 2 m of fallow land to prevent the lateral movement of water from one plot to another. Furthermore, the sub-plots within each irrigation treatment were isolated by 0.7 m of fallow land, and each experimental plot was 11.2 m^2^ (16 × 0.7 m).

#### 2.2.1. Mulching Treatments

Five mulching treatments were designed: control (without mulch (WM)), and four different surface mulching treatments, namely, poultry manure mulch (PMM), rice straw mulch (RSM), white plastic mulch (WPM), and black plastic mulch (BPM). Chemical analysis of PMM and RSM are presented in Table 3.

#### 2.2.2. Irrigation Treatments

Three irrigation treatments were applied as a percentage of the crop evapotranspiration (ET_c_) based on the evaporation pan method, and those treatments including I_100%_ (100% of ET_c_), I_85%_ (80% of ET_c_), and I_60%_ (60% of ET_c_).

### 2.3. Irrigation Water Applied (IWA)

The FAO Class A pan method was used to compute the reference evapotranspiration [20]:*ET_o_* = *K_pan_*× *E_pan_*(1)
where *E_pan_* is evaporation from the pan, mm day^−1^; *K_pan_* is the pan evaporation coefficient, 0.80 [20].

Maximum and minimum air temperatures, mean relative humidity, wind speed, and evaporation from Class A pan are shown in Table 4. Potato crop evapotranspiration and irrigation water applied for two successive years (2016 and 2017) are shown in Table 5.

The crop evapotranspiration (ET_c_) is then calculated using Equation (2):*ET_c_* = *ET_o_* × *K_c_*(2)
where *ET*_c_ is crop evapotranspiration, mm day^−1^; *K*_c_ is crop coefficient.

The lengths of various crop growth stages were 25, 30, 45, and 30 days for initial, crop development, mid-season, and late-season stages, respectively. The potato crop coefficients of initial, mid, and end stages were 0.50, 1.15, and 0.75, respectively [21].

The irrigation water applied (IWA) to every treatment was determined using Equation (3):(3)$$\mathrm{IWA}=\frac{A\times ETc\times Ii\times Kr}{Ea\times 1000}$$
where IWA is irrigation water applied, m^3^; A is area of plot, m^2^; Ii is irrigation intervals, day; K_r_ is coverage coefficient (Kr = (0.10 + G_c_) ≤ 1) [22]; G_c_ is ground cover; E_a_ is application efficiency (85%).

The quantities of applied irrigation water were 4970 and 5159 m^3^ ha^−1^ for two successive years, 2016 and 2017, respectively (Table 5). The irrigation treatments started directly after full emergence, based on scheduled irrigation treatments. The electrical conductivity of the used irrigation water (ECi) was 0.9 dS m^−1^.

The potato seeds tubers (Spunta) were planted by hand on 9 February during both growing seasons. Before planting, the soil was plowed, leveled, and then furrowed at appropriate distances to suit the crop in the trial. All mulch types were laid out after soil preparation and before planting. A polyethylene mulch having holes of 50 mm diameter at a distance of 30 cm was spread over the beds. The tuber seeds were then buried into the soil through the polyethylene holes. Meanwhile, the mulch remained on the soil surface till the end of the season. All other recommended cultural practices were then carried out in the experimental unit during both seasons.

### 2.4. Soil Sampling

Soil samples were collected twice: in the first season (before plant seeding) at three depths of 0–20, 20–40, and 40–60 cm (Table 2) and at the final harvest (after the second season), using an auger from the boreholes for moisture and salinity measurements. The samples were taken at 20 cm intervals from the soil surface to 60 cm deep at one position (0–20, 20–40, and 40–60 cm), and at 10 cm distance from the dripper to 30 cm (10, 20, and 30 cm). Approximately 500 g of soil was obtained for each sample to be tested. The soil samples were divided into two parts: one used to measure soil–water content by the gravimetric method [23], and one used to prepare soil solutions. The samples were crushed and passed through a 2.0-mm sieve after drying at 105 °C for 8 h. The EC of the paste extracts was then measured with a soil/water ratio of 1:5 (weight), using a conductivity meter (model 3200, YSI, Inc., Yellow Springs, OH, USA).

### 2.5. Vegetative Growth Parameters and Potato Yield

During harvest, five random plants were selected in each experimental unit to determine the vegetative growth parameters (stem length and the number of branches per plant) and tuber yield. Hand-harvested potato tuber yield (t ha^−1^) was measured, and sorted into three fractions based on tuber diameter, such as A grade: >60 mm, Mini tuber-1 (M-1): 30–60 mm, and Mini tuber-2 (M-2): <30 mm.

### 2.6. Crop Water Productivity (CWP)

The CWP values were calculated for every treatment after harvest using Equation (4) [24]:(4)$$\mathrm{CWP}=\frac{Potato\;tuber\;yield\;\left(kg\cdot ha^{-1}\right)}{Irrigation\;water\;applied\;\left(m^{3}\cdot ha^{-1}\right)}$$

### 2.7. Yield Response Factor (K_y_)

K_y_ was calculated using Equation (5) [25]:(5)$$\left(1-\frac{Ya}{Ym}\right)=K_{y}\left(1-\frac{ETa}{ETm}\right)$$
where Ya: potato crop of deficit irrigated treatments (I_80%_ and I_60%_ of ETc), t ha^−1^;

  Ym: potato crop of I_100%_ of ETc treatment, t ha^−1^; 

  ETa: actual crop evapotranspiration for the deficit treatments, mm;

  ETm: maximum crop evapotranspiration for the fully irrigated treatment, mm.

### 2.8. Statistical Analysis

All data were subjected to analysis of variance (ANOVA) for a strip-plot system in a randomized complete block design with three replications after testing for homogeneity of error variances, according to the procedure outlined by Gomez and Gomez [26], using InfoStat [27]. Significant differences between treatments were compared at *p* ≤ 0.05 using Duncan’s multiple range tests.

## 3. Results and Discussion

### 3.1. Potato Yield and Its Components

The plant height, number of branches, and yield of potato were positively influenced by the irrigation and mulching treatments. All the parameters significantly increased with mulching treatments over WM treatments (Table 6). The average potato tuber yield increased due to mulching treatments of PPM, RSM, WPM, and BPM over WM treatment by 75.26%, 56.77%, 45.11%, and 33.03% in 2016, and by 65.74%, 50.31%, 37.42%, and 22.53% in 2017, respectively. These results demonstrated that mulching could mitigate the effects of water stress on plant growth and produce optimal potato yields. The yield increased due to the applied mulching, which can be attributed to a lower rate of water loss from the soil by evaporation, leading to significant conservation of soil moisture. Hence, more water became available for the potato crop, thereby decreasing the salt content in the soil surface and increasing the crop yield [17]. Moreover, the applied mulch increased transpiration, thereby leading to more photosynthetic efficiency that resulted in increased yield [28]. Furthermore, soil moisture changes in the upper surface layer (0–10 cm) were highly dynamic due to water vapor fluxes across the soil–atmospheric interface [29]. However, mulching reduced the fluctuation of soil moisture and soil temperature [30]. Thus, the fluctuation of soil moisture, especially in upper soil layers, has been reported to negatively influence seed germination and plant growth [31].

Considering the effect of different mulching materials on the average potato tuber yield, the results indicated that organic mulches (PMM and RSM) increased the average potato tuber yield (40.17 and 36.19 t ha^−1^, respectively) compared to inorganic mulches (WPM and BPM; 33.28 and 30.10 t ha^−1^), respectively (Table 6). This could be due to the enhanced availability and release of nutrients from decomposed organic mulches, improved soil physical properties, increased soil–water holding capacity leading to good aeration and drainage, better root growth, and enhanced nutrient absorption by crop plants [32]. Moreover, these results can be attributed to the diminishing of water loss from the soil surface through evaporation for inorganic mulch compared to organic mulch. This resulted in poor aeration with a high moisture regime for inorganic mulch, thereby drastically lowering tuber yield [33]. In addition, Wan and Kang [34] stated that excess water in the soil decreased the oxygen diffusion rate in the root zone, which negatively affected crop yield.

Table 6 indicates that all parameters were significantly affected by irrigation treatments. The average potato tuber yield increased with increasing the quantity of applied irrigation water. In 2016, there was an increase in the average potato tuber yield by 13.83% and 29.16% owing to I_100%_ treatment in comparison with I_80%_ and I_60%_, respectively. However, the corresponding values were 12.95% and 30.91% in 2017, respectively. These results agree with the findings of Abd El-Wahed and Ali [35], who reported that optimal irrigation produced the highest yield due to using the highest amount of water. This was due to the sufficient available water in the root zone for the I_100%_ treatment, thereby leading to an increase in both water and nutrient absorption. Consequently, an increase in plant metabolic mechanisms increased potato tuber yields. On the other hand, the yield was significantly decreased when there was insufficient available water in the root zone (I_60%_), which negatively affected cell division and expansion [36]. The limited tolerance of potato to drought is due to its relatively shallow root system, and the tendency of the stomata to become closed under water-limited conditions, thereby reducing leaf extension rates [37]. Stomatal closure also reduced the uptake of CO_2_, and as a result, photosynthetic activity was decreased, leading to a substantial rise in leaf temperature and photorespiration [38].

The interaction effect of irrigation and mulching significantly influenced the plant height, number of branches, and tuber yield of potato (Table 6). Moreover, the yield was significantly affected by the interaction between mulching materials and irrigation treatments. The highest potato tuber yield of 44.91 and 45.79 t ha^−^^1^ was recorded with the interaction of I_100%_ treatment and applied PMM in 2016 and 2017, respectively. However, the lowest potato tuber yield (18.64 and 20.59 t ha^−^^1^) was obtained from plants subjected to the I_60%_ treatment and WM during both seasons, respectively (Table 6). Furthermore, the potato tuber yields of 39.17 and 41.22 t ha^−^^1^ were recorded for the I_80%_ treatment and PPM during 2016 and 2017, respectively. Therefore, it was observed that the water discharge to satisfy the crop water demand, full irrigation (I_100%_), and PMM treatment remained unmatched by producing the maximum potato tuber yield with minimal saline toxicity. For insufficient discharge, the deficit irrigation (I_80%_) and PMM treatment could be recommended to conserve 20% of the irrigation water, thereby producing a significant potato tuber yield but with a slight increase of soil salinity.

Table 6 shows that the I_60%_ treatment with mulch produced the highest potato tuber yields, as compared to I_80%_ and I_100%_ treatments without mulch. The increased yield for the mulch with a lower water regime (I_60%_) may be due to better water utilization, higher nutrient uptake, and excellent soil–water–plant relationships. This result agrees with the result of Biswas et al. [33], who reported that drip irrigation at 50% with mulch produced the highest tomato yield, as compared to 75% and 100% irrigation without mulch.

The size of the potato tuber was significantly affected by mulching materials (Table 7). The highest weight of an A-grade tuber (>60 cm) was recorded for PMM, which significantly differed from the rest of the mulching treatments. Moreover, the average weight of A-grade tuber values of mulching treatments (PPM, RSM, WPM, and BPM), in comparison with the values of WM treatment, increased by 75.55%, 68.11%, 62.41%, and 54.90% in 2016 and by 74.14%, 66.73%, 60.31%, and 51.03% in 2017, respectively. These results indicated that the size of tubers, considering the mulching materials, were, in decreasing order, RSM > WPM > BPM.

In addition, organic mulches (PMM and RSM) increased the average yield of 28.13 and 21.71 t ha^−1^ of A-grade tubers in comparison to the inorganic mulches (WPM and BPM) of 18.30 and 15.04 t ha^−1^ during the two seasons, respectively. A similar trend was recorded by Dvořák et al. [39], who reported that the use of organic mulch (chopped grass) resulted in a significant increase in the weight of tubers (tuber fraction 56–60 mm and >60 mm) compared to inorganic mulch (black mulch).

Furthermore, Table 7 shows that the sizes of the potato tuber yield were significantly affected by the irrigation treatments. The maximum values of the tuber yield size were obtained at I_100%_ irrigation level. As an average, the maximum values of A-grade tubers (˃ 60 cm; 20.04 and 20.63 t ha^−1^) were obtained for plants irrigated with the highest level (I_100%_) during two seasons, respectively. Moreover, the minimum values (15.66 and 15.87 t ha^−1^) were obtained for the lowest irrigation level (I_60%_) during both seasons, respectively. The reduction in A-grade tuber yield for the I_60%_ treatment was generally attributed to soil moisture reduction at the tuber initiation stage, which has been described as one of the critical growth stages of potato. The yield reduction for deficit irrigation was generally attributed to a large reduction in the weight of tubers (> 0.227 kg/tuber) [40]. Meanwhile, Shock et al. [41] evaluated potato cultivars at deficit irrigation levels and concluded that most of the cultivars showed a significant difference in the total or numbers of A-grade tuber yield under water-limited conditions. It was also inferred that potato genotypes differed significantly in water use efficiency, which determined crop growth and tuber yield.

It is pertinent to state that the interaction effect of mulching and irrigation treatments on the size of potato tuber yield was significant during both experimental seasons. The highest values of A graded tubers (>60 cm; 31.44 and 32.06 t ha^−1^ during 2016 and 2017, respectively) were obtained for plants irrigated with the highest level (I_100%_) and applied PMM. In contrast, the lowest values of A-grade tubers (>60 cm; 5.59 and 6.19 t ha^−1^ during 2016 and 2017, respectively) were obtained for plants irrigated with the lowest level (I_60%_) of irrigation and WM (Table 7).

### 3.2. Crop Water Productivity (CWP)

CWP values were significantly affected by mulching and irrigation treatments (Figure 1). However, during both crop growing seasons, the highest CWP values (10.20 and 10.06 kg m^−3^) were recorded for PMM followed by RSM (9.16 and 9.15 kg m^−3^), while WM treatments (5.76 and 6.03 kg m^−3^) remained inferior to all mulches. This might be attributed to better tuber yields obtained for PMM treatment (39.67 and 40.67 t ha^−1^) than the corresponding tuber yields obtained for other treatments with the same amount of irrigation water applied in both seasons. Moreover, this result was due to reduced soil evaporation and increased plant transpiration [42].

On the other hand, Figure 1 demonstrates that the CWP was significantly affected by the irrigation treatments. The highest CWP values of 9.44 and 9.27 kg m^−3^ were obtained for the I_60%_ treatment compared to the I_100%_ treatment (7.16 and 7.28 kg m^−3^) in the two seasons, respectively. These results agree with the results of Abd El-Wahed et al. [17].

Nonetheless, the highest CWP values (11.71 and 11.31 kg m^−3^) were recorded for potato plants irrigated with the lowest level (I_60%_) and PMM in 2016 and 2017, respectively. In contrast, lowest CWP values (5.50 and 5.69 kg m^−3^) were obtained for potato plants irrigated with the highest level (I_100%_) and WM in both seasons, respectively. These results are due to decreased water applied for I_60%_ in comparison with other treatments (I_100%_ and I_80%_). This result agrees with the result of Biswas et al. [33], who reported that decreased irrigation water with mulch increased the CWP.

### 3.3. Yield Response Factor (Ky)

K_y_ indicates a linear relationship between the reduction in relative yield based on the reduction in irrigation water applied, as shown in Figure 2. This result agrees with the results of Lovelli et al. [43], who reported that K_y_ usually indicates a linear relationship of the relative reduction in water that is consumed with a relative reduction in yield.

Figure 2 shows that the yield response factor for the potato crop was 0.58 and 0.59 during 2016 and 2017, respectively, thereby indicating that the potato is very sensitive to water deficit. Doorenbos and Kassam [44] stated that when K_y_ < 1, the yield loss is less important than the evapotranspiration deficit. Meanwhile, when K_y_ > 1, the yield loss is more important than the evapotranspiration deficit, while the yield loss is equal to the evapotranspiration deficit when K_y_ = 1.

### 3.4. Distribution of Salt

The soil salinity was affected by mulching materials and irrigation treatments (Figure 3). Moreover, there were differences between initial soil ECe (measured before c seeding) and after all treatments (at harvest; Table 2).

For all irrigation treatments, the ECe increased with increased distance from the drippers toward the fringes of the irrigated area. The soil salts migrated away with irrigation water around the drippers in all directions, such that the salts in the soil surface were leached during the process [45]. Thus, low salinities were in close proximity to the dripper, thereby providing a zone of decreased osmotic potential that reduces the osmotic stress on plant growth [46].

In addition, the least irrigation (I_60%_) generally resulted in greater salt content of the upper soil layer, while the lowest ECe values were obtained with full irrigation (I_100%_; Figure 3). The average soil salinity of the I_60%_ treatment remained the highest (16.51% and 9.79%) compared to the average soil salinity of the I_100%_ and I_80%_ treatments, respectively. These results might be linked to more available water in the root zone for the I_100%_ treatment compared to the I_60%_ treatment, thereby decreasing soil salinity in the upper soil layer.

Figure 3 shows the ECe accumulation in the 0–60 cm soil layer. Irrigation treatment at I_60%_ resulted in the least soil–water content and the highest ECe value, which negatively affected potato growth and significantly reduced potato tuber yield (28.16 and 28.70 t ha^−1^) in the 2016 and 2017 seasons, respectively. In contrast, the highest IWA values (I_100%_) always resulted in the highest soil–water content and lowest ECe, thereby resulting in the highest potato tuber yield (36.37 and 37.58 t ha^−1^) in the 2016 and 2017 seasons, respectively.

These findings are in confirmation with those of Abd El-Mageed et al. [47], who reported that all mulching materials effectively reduced salt accumulation in the root zone by reducing the flux of water from lower to higher soil horizons. The average ECe value of WM was increased by 10.05%, 16.08%, 21.59%, and 18.54% compared to the corresponding values of PMM, RSM, WPM, and BPM, respectively. These findings could be attributed to lower concentration of soluble salts applied with irrigation water (0.9 dS m^−1^), WM treatment had higher water evaporation from the upper soil layer, which allowed a greater upward movement of salt from the deeper soil layers to the upper soil layer, and mulching reduced soil water loss by evaporation from the soil surface. The EC of soil under mulching treatment was ~38% less in comparison with no mulch (control) [48]. Mulching conserves moisture and facilitates its infiltration into the soil profile [49], increasing available soil water [50] and facilitates the leaching of salts accumulated from the upper soil layer to the deeper soil layers, thereby decreasing soil salinity in the upper soil layer.

Moreover, Figure 3 shows that the average ECe values during the two experimental seasons were affected by the mulching materials in the order of PMM > RSM > BPM > WPM. These results are consistent with the findings of [51]. Plastic mulch treatment conserved the highest amount of soil moisture in comparison to organic mulch treatments. Meanwhile, both treatments conserved greater moisture than the bare soil. Hence, the use of mulch in conjunction with dripper irrigation would decrease salt accumulation at the soil surface by reducing direct evaporation of soil moisture.

## 4. Conclusions

Our results indicate that the highest values of potato tuber yield were achieved by I_100%_ in comparison to I_80%_ and I_60%_, while PMM remained superior in comparison to the rest of the mulching treatments. The highest salt accumulation in the soil profile was observed under I_60%_ and WM treatments. Under the environmental conditions of Egypt, when the water discharge is sufficient, full irrigation (I_100%_) and PMM could be recommended to achieve the maximum potato tuber yield without increasing soil salinity. Overall, when the water discharge is insufficient, deficit irrigation (I_80%_) and PMM could serve as a potential approach to conserve 20% of irrigation water, with a minimum reduction in tuber yields and a slight increase in soil salinity. However, further research is suggested to evaluate the productivity of drought-resistant genotypes of potato along with field evaluations of other mulching materials especially from biotic origins.

## Figures and Tables

**Figure 1 plants-09-00701-f001:**
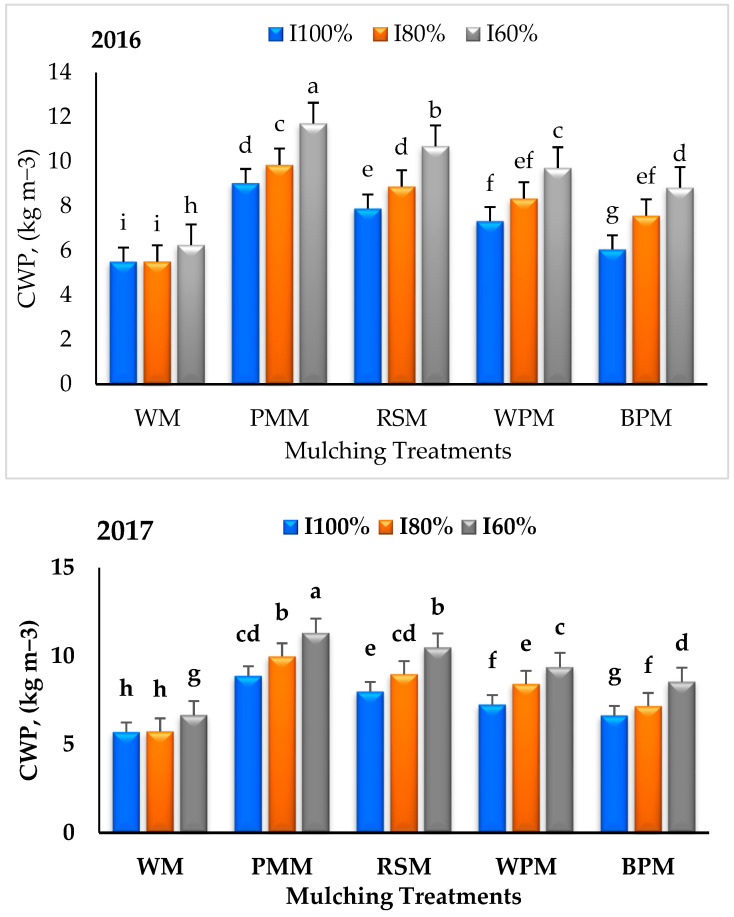
Effect of irrigation and mulching treatments on the CWP in 2016 and 2017. Means have different letters are significantly different at 5%.

**Figure 2 plants-09-00701-f002:**
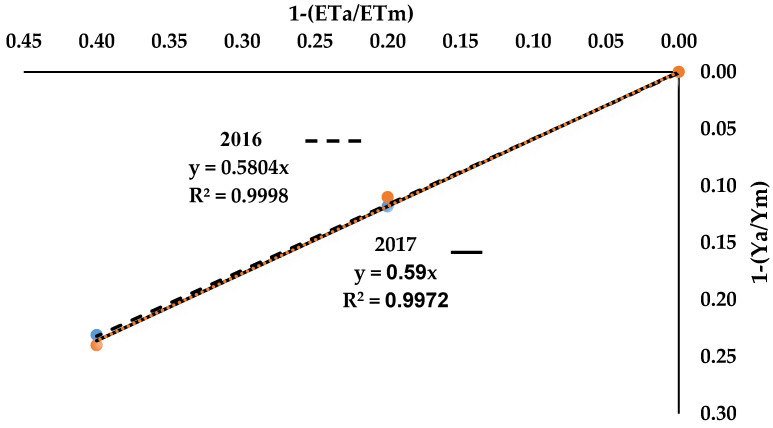
Yield response factor of potato yield under irrigation treatments during 2016 and 2017.

**Figure 3 plants-09-00701-f003:**
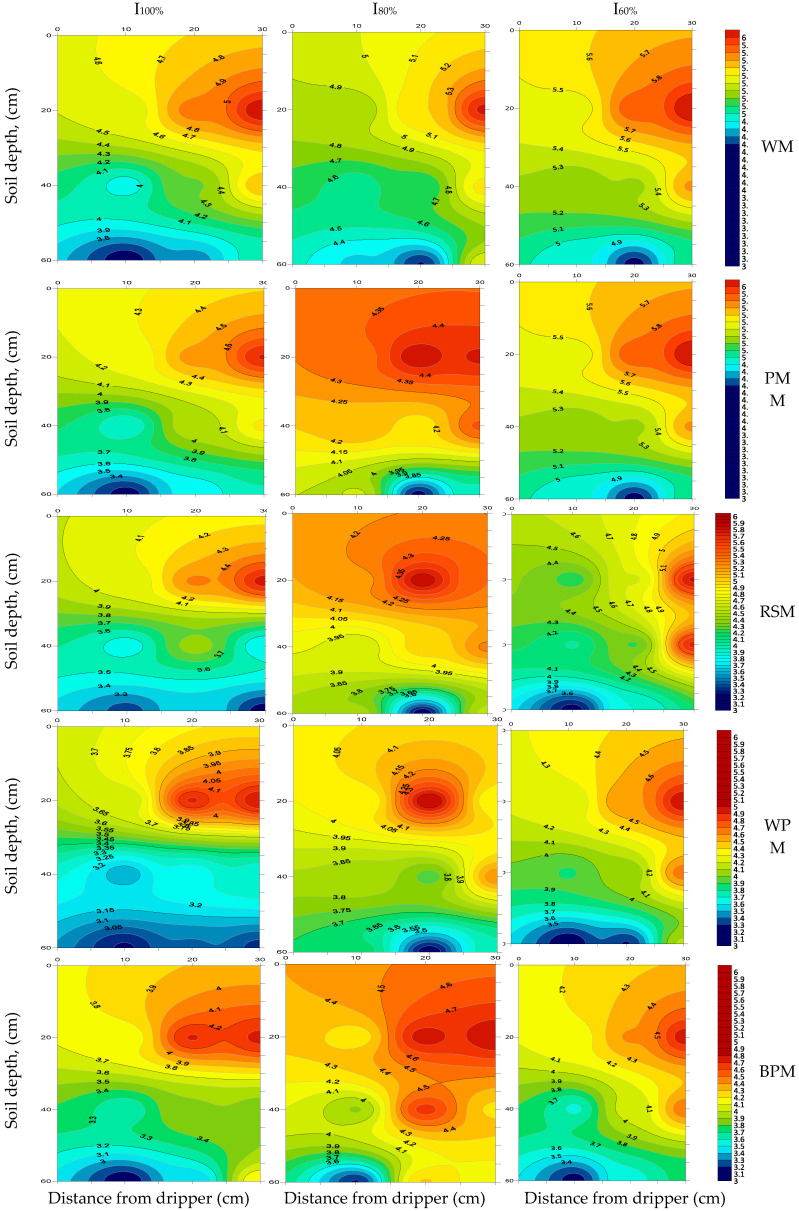
Distribution of ECe in the 0–60 cm soil depth and around the dripper under treatments.

**Table 1 plants-09-00701-t001:** Some physical properties of the experimental soil.

Soil Depth, cm	Particle Size Distribution				ρb, Mg m^−3^	FC %	WP %	AW %
	Sand, %	Silt, %	Clay, %	Texture Class				
0–20	72.50	12.90	14.60	SL	1.46	19.79	5.69	14.10
20–40	74.60	12.00	13.40	SL	1.57	17.42	3.64	13.78
40–60	74.20	12.10	13.70	SL	1.58	18.62	4.37	14.25

SL: sandy loam, ρb: bulk density, FC: field capacity, WP: wilting point, and AW: available water.

**Table 2 plants-09-00701-t002:** Some chemical properties of the experimental soil.

Soil Depth, cm	ECe, dSm^−1^	pH	CaCO_3_ %	OM *	Total N mg kg^−1^	Available Nutrients (mg kg^−1^)				
						P	K	Fe	Mn	Zn
0–20	4.70	7.86	7.5	0.90	14.81	3.25	42.57	4.45	1.25	0.87
20–40	4.40	7.78	6.6	0.82	13.24	3.28	39.87	4.35	1.08	0.79
40–60	4.10	7.92	7.4	0.65	13.21	3.15	39.24	4.21	0.88	0.82

* ECe: electrical conductivity; OM: organic matter content; P: extracted with NaHCO_3_, pH: 8.5; K: extracted with ammonium acetate, pH7. Fe, Mn, Zn, P: extracted with diethylene triamine pentaacetic acid (DTPA).

**Table 3 plants-09-00701-t003:** Chemical properties of poultry manure mulch and rice straw mulch.

Properties	PMM	RSM
Total N (%)	1.65	0.54
Total P (%)	2.4	0.101
Total (K%)	2.1	0.379
OM (%)	37.2	74.0
C (%)	32.3	42.9
C/N ratio	19.6:1	79:1

**Table 4 plants-09-00701-t004:** Maximum and minimum air temperature, mean relative humidity, wind speed, and evaporation from Class A pan for two successive years (2016 and 2017).

Month	Tmin	Tmax	RH	Ws	Epan
	(°C)	(°C)	(%)	(m s^−1^)	(mm d^−1^)
2016					
February	9.9	25.8	59.2	2.7	2.37
March	13.4	28.0	56.7	3.2	3.74
April	17.0	36.1	53.2	4.0	5.22
May	19.8	36.0	51.8	4.6	6.94
June	24.3	37.3	54.1	4.5	7.65
2017					
February	10.6	26.1	56.9	2.5	2.54
March	13.9	28.9	57.9	3.8	3.87
April	18.919.5	35.2	56.0	4.2	5.66
May	19.3	36.5	52.7	4.9	7.03
June	24.919.5	36.4	50.7	4.7	7.68

**Table 5 plants-09-00701-t005:** Potato crop evapotranspiration and irrigation water applied for two successive years (2016 and 2017).

Months	Decade	Stage	Kc	ETo (mm d^−1^)		ETc (mm d^−1^)		IWA m^3^ ha^−1^	
				2016	2017	2016	2017	2016	2017
**February**	2	Init.	0.5	1.9	2.03	0.95	1.02	13.41	14.33
	3	Init.	0.5	1.9	2.03	0.95	1.02	26.82	28.66
**March**	1	Init./Dev.	0.55	2.99	3.1	1.64	2.48	67.71	102.12
	2	Dev.	0.8	2.99	3.1	2.39	2.48	132.26	137.13
	3	Dev.	1	2.99	3.1	2.99	3.10	207.54	215.18
**April**	1	Dev./Mid	1.1	4.18	4.53	4.6	4.98	378.66	410.36
	2	Mid	1.15	4.18	4.53	4.81	5.21	463.73	502.56
	3	Mid	1.15	4.18	4.53	4.81	5.21	463.73	502.56
**May**	1	Mid	1.15	5.55	5.62	6.38	6.46	705.83	714.73
	2	Mid	1.15	5.55	5.62	6.38	6.46	705.83	714.73
	3	Late	1	5.55	5.62	5.55	5.62	652.94	661.18
**June**	1	Late	0.85	6.12	6.14	5.2	5.22	612.00	614.00
	2	Late	0.75	6.12	6.14	4.59	4.61	540.00	541.76
								4970.48	5159.31

**Table 6 plants-09-00701-t006:** Effect of irrigation and mulching treatments and their interaction on the plant height, number of branches, and total yield during 2016 and 2017.

Treatments		Plant Height (cm)		Number of Branches		Total Yield (t ha^−1^)	
		2016	2017	2016	2017	2016	2017
I_100%_		61.79 ^A^	61.96 ^A^	2.73 ^A^	3.20 ^A^	36.37 ^A^	37.58 ^A^
I_80%_		55.69 ^B^	55.55 ^B^	2.67 ^AB^	3.20 ^A^	31.94 ^B^	33.27 ^B^
I_60%_		48.51 ^C^	49.09 ^C^	2.00 ^B^	2.40 ^A^	28.16 ^C^	28.70 ^C^
WM		46.87 ^D^	47.01 ^D^	2.00 ^B^	2.11 ^C^	22.63 ^E^	24.54 ^E^
PMM		60.97 ^A^	60.88 ^A^	2.78 ^A^	3.11 ^AB^	39.67 ^A^	40.67 ^A^
RSM		58.76 ^B^	59.00 ^AB^	2.78 ^A^	3.33 ^A^	35.49 ^B^	36.89 ^B^
WPM		55.90 ^C^	56.39 ^BC^	2.67 ^AB^	3.33 ^A^	32.84 ^C^	33.72 ^C^
BPM		54.17 ^C^	54.38 ^D^	2.11 ^AB^	2.78 ^B^	30.11 ^D^	30.08 ^D^
I_100%_	WM	53.20 ^g^	53.87 ^e^	4.13 ^cd^	4.17 ^fg^	27.35 ^hi^	29.35 ^e^
	PMM	67.03 ^a^	66.90 ^a^	4.57 ^abc^	4.8 ^bc^	44.91 ^a^	45.79 ^a^
	RSM	64.67 ^ab^	64.70 ^ab^	5.03 ^a^	5.10 ^a^	39.22 ^b^	41.20 ^b^
	WPM	62.27 ^b^	62.80 ^abc^	4.37 ^bcd^	4.57 ^cd^	36.41 ^c^	37.39 ^c^
	BPM	61.80 ^bc^	61.53 ^bcd^	4.17 ^cd^	4.27 ^efg^	33.89 ^de^	34.18 ^d^
I_80%_	WM	46.80 ^h^	45.87 ^fg^	4.07 ^cd^	4.03 ^gh^	21.92 ^j^	23.68 ^g^
	PMM	61.00 ^bcd^	61.17 ^bcd^	4.37 ^bcd^	4.50 ^de^	39.17 ^b^	41.22 ^b^
	RSM	57.63 ^def^	57.90 ^de^	4.83 ^ab^	4.90 ^ab^	35.32 ^cd^	37.03 ^c^
	WPM	58.33 ^cde^	58.27 ^cde^	4.23 ^bcd^	4.30 ^defg^	33.18 ^de^	34.77 ^cd^
	BPM	54.67 ^efg^	54.53 ^e^	4.03 ^cd^	4.10 ^gh^	30.11 ^fg^	29.61 ^e^
I_60%_	WM	40.60 ^i^	41.30 ^g^	3.77 ^d^	3.87 ^h^	18.64 ^k^	20.59 ^h^
	PMM	54.87 ^efg^	54.57 ^e^	4.07 ^cd^	4.43 ^def^	34.92 ^cd^	35.03 ^cd^
	RSM	53.97 ^fg^	54.40 ^e^	4.30 ^bcd^	4.50 ^de^	31.89 ^ef^	32.44 ^d^
	WPM	47.10 ^h^	48.10 ^f^	3.90 ^d^	4.27 ^efg^	28.96 ^gh^	29.01 ^ef^
	BPM	46.03 ^h^	47.07 ^f^	3.83 ^d^	4.03 ^gh^	26.32 ^i^	26.44 ^f^

Means have different letters are significantly different at 5%.

**Table 7 plants-09-00701-t007:** Effect of irrigation and mulching treatments and their interaction on the size of potato tubers during 2016 and 2017.

Treatments		A Grade (>60 mm)		M-1 (30–60 mm)		M-2 (˂30 mm)	
		2016	2017	2016	2017	2016	2017
I_100%_		20.04 ^A^	20.63 ^A^	10.71 ^A^	11.07 ^A^	5.62 ^A^	5.86 ^A^
I_80%_		17.71 ^B^	18.42 ^B^	9.42 ^B^	9.81 ^B^	4.81 ^B^	5.05 ^B^
I_60%_		15.66 ^C^	15.87 ^C^	8.31 ^C^	8.47 ^C^	4.19 ^C^	4.38 ^C^
WM		6.78 ^E^	7.35 ^E^	6.78 ^D^	7.35 ^E^	9.04 ^A^	9.81 ^A^
PMM		27.77 ^A^	28.49 ^A^	7.93 ^C^	8.14 ^D^	3.98 ^C^	4.07 ^C^
RSM		21.30 ^B^	22.13 ^B^	10.64 ^B^	11.07 ^B^	3.55 ^D^	3.69 ^D^
WPM		18.06 ^C^	18.54 ^C^	11.50 ^A^	11.80 ^A^	3.28 ^D^	3.38 ^E^
BPM		15.04 ^D^	15.04 ^D^	10.54 ^B^	10.52 ^C^	4.52 ^B^	4.50 ^B^
I_100%_	WM	8.21 ^i^	8.81 ^i^	8.21 ^fg^	8.81 ^hi^	10.95 ^a^	11.73 ^a^
	PMM	31.44 ^a^	32.06 ^a^	8.97 ^ef^	9.16 ^ghi^	4.50 ^ef^	4.57 ^e^
	RSM	23.54 ^c^	24.73 ^c^	11.78 ^b^	12.35 ^ab^	3.93 ^g^	4.12 ^fgh^
	WPM	20.02 ^de^	20.56 ^e^	12.73 ^a^	13.09 ^a^	3.64 ^gh^	3.74 ^ij^
	BPM	16.95 ^g^	17.09 ^f^	11.85 ^b^	11.97 ^bc^	5.09 ^d^	5.12 ^d^
I_80%_	WM	6.57 ^j^	7.12 ^j^	6.57 ^h^	7.12 ^j^	8.76 ^b^	9.47 ^b^
	PMM	27.44 ^b^	28.87 ^b^	7.85 ^g^	8.26 ^i^	3.927 ^g^	4.12 ^fg^
	RSM	21.21 ^d^	22.23 ^d^	10.59 ^c^	11.11 ^cd^	3.52 ^gh^	3.71 ^ij^
	WPM	18.25 ^f^	19.11 ^e^	11.61 ^b^	12.16 ^b^	3.33 ^h^	3.21 ^jk^
	BPM	15.04 ^h^	14.80 ^g^	10.54 ^c^	10.35 ^de^	4.52 ^e^	4.43 ^ef^
I_60%_	WM	5.59 ^k^	6.19 ^j^	5.59 ^i^	6.19 ^k^	7.45 ^c^	8.23 ^c^
	PMM	24.44 ^c^	24.51 ^c^	6.97 ^h^	7.00 ^jk^	3.50 ^gh^	3.50 ^jk^
	RSM	19.15 ^ef^	19.47 ^e^	9.57 ^de^	9.73 ^efg^	3.19 ^hi^	3.24 ^kl^
	WPM	15.92 ^gh^	15.97 ^fg^	10.14 ^cd^	10.16 ^ef^	2.90 ^i^	2.90 ^l^
	BPM	13.16 ^i^	13.23 ^h^	9.21 ^e^	9.26f ^gi^	3.95 ^fg^	3.97 ^ghi^

Means have different letters are significantly different at 5%.

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
