# Peer review of "Salt Distribution and Potato Response to Irrigation Regimes under Varying Mulching Materials"

_plants, 2020, doi:10.3390/plants9060701_

Round 1

Reviewer 1 Report

The research aims to develop deficit (trickle) irrigation and mulching strategies to enhance potato crop (tuber) water productivity/yield stability. Results obtained from two years of field studies at a university experimental station, which presumably is typical of agricultural crop growing conditions locally, were the basis of data analysis, conclusions, and interpretations. Poultry manure mulch facilitated higher potato tuber yield under full and 80% of full irrigation. 

The research appeared robust with a comprehensive experimental plan and experimentation, and thorough data collection and analysis, including statistical analyses. All parts of the process were explained in detail and seemed thoroughly executed. It is difficult to assess technical efficacy completely as numerous procedures were involved but there were no obvious reasons for concern. The authors should emphasize the principal knowledge gap/discovery addressed by the research and resolved by the results.

The text is procedurally comprehensive. Experiments and techniques are thoroughly described, and results presented in detail. Having said this, the text is difficult to follow to the extent it adversely affects the overall communication. From a stylistic perspective, it needs a succinct declarative presentation style with precise word usage generally and technically. Clear strategic decisions should plan and carry out what is to be communicated in each principal paper section, e.g., title, abstract, etc. The text contains numerous acronyms (too many) to the point that these reduce communication effectiveness for all but especially the broader journal readership. The journal readership will be beyond a narrow topic focus group and the text must be written for them as well. Specific comments are listed below. Minimize the overuse of acronyms by explaining in words critical information and interpretations.  

Specific comments and queries:

Title - Write a declarative phrase that captures critical results and discoveries. In the present form, keywords are listed but do not coalesce well into an effective phrase exemplifying the research accomplishments.

Abstract - Rewrite the text using sentences that interlink the entirety of what is trying to be communicated, i.e., transitions are too abrupt. For example, the first sentence (introducing the problem) does not transition clearly to the second, which is about what/how research/results in this study add to important problem solutions. Lines 18 to 22 are too technical (with too many acronyms) to explain the experimental approach. Try a more generic statement of approach that not only introduces but also rationalizes it well. The sentence (about critical results) in lines 22 and 23 needs to be more precise, i.e. what was the principal research discovery that warrants first mention. Thereafter, eliminate the acronyms and make declarative/definitive statements of critical conclusions based on the results. Add a closing sentence emphasizing the principal discovery(ies) and future directions. This text and the title are two of the most important sections. Please use the text edits suggested here when revising the remainder of the paper sections.     

Introduction - The necessary sections appear to be present. Suggestions are to write the text to highlight critical knowledge gaps to be addressed by the research and add critical results statements in the last paragraph.        

Materials and Methods - This section is comprehensive and informative. Try to make it clear to a broader journal readership. Tables 1, 2, and 3 legends are insufficient. The legends should as stand-alone as possible. More detailed legends will help communicate the methods more clearly. Consider an introductory Research section where these tables might be included instead of referencing these in the Methods. 

Results and Discussion: 

The datasets are comprehensive, informative, and robust. Consequently, there are not critical concerns but there are stylistic suggestions that are meant to improve communication effectiveness and for emphasizing critical results/discovery(ies).

Separate the text into separate Results and Discussion sections. The integration is not communicating well.

Legends should be more detailed and self-explanatory, i.e., making evident what was done and why.

The legend should include a highlighted title that, like the paper title, makes evident critical dataset interpretations.

Results sections titles should be declarative stating why the research was done, presenting critical results, and what these mean, and emphasizing novel conclusions. When presenting critical results, it will be possible also to include the necessary technical details. Section titles should indicate the experimental rationale.  

Reduce text that restates what is evident in the dataset illustrations. Restrict it to focus on critical results and interpretations. Restrict acronym use but, if necessary, make sure to re-introduce the meaning when an acronym is first used in this section. Multi paragraph sections should include a one-sentence statement of conclusion, i.e. makes evident how the experiments/results are linked.

Discussion - The first paragraph summarizes pre-research critical gaps of knowledge, research results that resolved these and why, and introduces principal discussion points (likely no more than two or three). Each discussion point should have a declarative title followed by text that emphasizes what the authors think is important based on the research results. Note, speculation about research results cause and effect, presently included in the Results w/reference citations, can be included here. A suggestion is to focus on results herein and speculate less about cause and effect for which there is little/no evidence for in this paper. The final paragraph (can be the Conclusion herein, stated more succinctly, declaratively, w/o acronyms) should also include a final sentence stating the overall research outcome and/or potential future critical research directions.

Author Response

ReviewerA:

The research aims to develop deficit (trickle) irrigation and mulching strategies to enhance potato crop (tuber) water productivity/yield stability. Results obtained from two years of field studies at a university experimental station, which presumably is typical of agricultural crop growing conditions locally, were the basis of data analysis, conclusions, and interpretations. Poultry manure mulch facilitated higher potato tuber yield under full and 80% of full irrigation.

Thank for the good comment

The research appeared robust with a comprehensive experimental plan and experimentation, and thorough data collection and analysis, including statistical analyses. All parts of the process were explained in detail and seemed thoroughly executed. It is difficult to assess technical efficacy completely as numerous procedures were involved but there were no obvious reasons for concern. The authors should emphasize the principal knowledge gap/discovery addressed by the research and resolved by the results.

Thank you so much for your informative suggestion, which have allowed us to make a considerable improvement of the manuscript

The text is procedurally comprehensive. Experiments and techniques are thoroughly described, and results presented in detail. Having said this, the text is difficult to follow to the extent it adversely affects the overall communication. From a stylistic perspective, it needs a succinct declarative presentation style with precise word usage generally and technically. Clear strategic decisions should plan and carry out what is to be communicated in each principal paper section, e.g., title, abstract, etc. The text contains numerous acronyms (too many) to the point that these reduce communication effectiveness for all but especially the broader journal readership. The journal readership will be beyond a narrow topic focus group and the text must be written for them as well. Specific comments are listed below. Minimize the overuse of acronyms by explaining in words critical information and interpretations.  

Thanks for good suggestion. We have done.

Specific comments and queries are:

  1. Title - Write a declarative phrase that captures critical results and discoveries. In the present form, keywords are listed but do not coalesce well into an effective phrase exemplifying the research accomplishments.

Response: Thanks for your suggestion. We have thoroughly checked and corrected.

  1. Abstract - Rewrite the text using sentences that interlink the entirety of what is trying to be communicated, i.e., transitions are too abrupt. For example, the first sentence (introducing the problem) does not transition clearly to the second, which is about what/how research/results in this study add to important problem solutions. Lines 18 to 22 are too technical (with too many acronyms) to explain the experimental approach. Try a more generic statement of approach that not only introduces but also rationalizes it well. The sentence (about critical results) in lines 22 and 23 needs to be more precise, i.e. what was the principal research discovery that warrants first mention. Thereafter, eliminate the acronyms and make declarative/definitive statements of critical conclusions based on the results. Add a closing sentence emphasizing the principal discovery(ies) and future directions. This text and the title are two of the most important sections. Please use the text edits suggested here when revising the remainder of the paper sections.

Response: Thanks, we have revised the manuscript according to your suggestion. we have revised after your comments.

3- Introduction - The necessary sections appear to be present. Suggestions are to write the text to highlight critical knowledge gaps to be addressed by the research and add critical results statements in the last paragraph.        

Response: The suggestion has been made

4- Materials and Methods - This section is comprehensive and informative. Try to make it clear to a broader journal readership. Tables 1, 2, and 3 legends are insufficient. The legends should as stand-alone as possible. More detailed legends will help communicate the methods more clearly. Consider an introductory Research section where these tables might be included instead of referencing these in the Methods.

We are happy to confirm that we have Modified.

5- Results and Discussion: The datasets are comprehensive, informative, and robust. Consequently, there are not critical concerns but there are stylistic suggestions that are meant to improve communication effectiveness and for emphasizing critical results/discovery(ies).

The suggestion has been added

6- Separate the text into separate Results and Discussion sections. The integration is not communicating well.

All suggestions have been modified in the text of the manuscript

7- Legends should be more detailed and self-explanatory, i.e., making evident what was done and why.

This suggestion is well taken

8- The legend should include a highlighted title that, like the paper title, makes evident critical dataset interpretations.

Corrected as suggested

9- Results sections titles should be declarative stating why the research was done, presenting critical results, and what these mean, and emphasizing novel conclusions. When presenting critical results, it will be possible also to include the necessary technical details. Section titles should indicate the experimental rationale.  

All suggestions have been modified

10- Reduce text that restates what is evident in the dataset illustrations. Restrict it to focus on critical results and interpretations. Restrict acronym use but, if necessary, make sure to re-introduce the meaning when an acronym is first used in this section. Multi paragraph sections should include a one-sentence statement of conclusion, i.e. makes evident how the experiments/results are linked.

Response: Thanks for your informative suggestion. We have thoroughly checked  the paper and revised accordingly.

11- Discussion - The first paragraph summarizes pre-research critical gaps of knowledge, research results that resolved these and why, and introduces principal discussion points (likely no more than two or three). Each discussion point should have a declarative title followed by text that emphasizes what the authors think is important based on the research results. Note, speculation about research results cause and effect, presently included in the Results w/reference citations, can be included here. A suggestion is to focus on results herein and speculate less about cause and effect for which there is little/no evidence for in this paper. The final paragraph (can be the Conclusion herein, stated more succinctly, declaratively, w/o acronyms) should also include a final sentence stating the overall research outcome and/or potential future critical research directions.

Response: The suggestion has been satisfied and we have checked and done

Reviewer 2 Report

The authors provided informative results on the effects of irrigation management practices on potato production in an arid region in Egypt, which looks to be an appealing research item for those in similar climates. However, the presentation of their methodologies and results is not as explicit as it is supposed to be. In the current form, the referee cannot recommend the manuscript for publication. The reasons for my conclusion follow below.  

First, it is not clear how the calculations were made with the information provided in the manuscript. In other words, the Materials and Methods section lacks details in the description. For example, how was the irrigation treatment applied? How often did the authors apply water for each treatment? When in a day did the authors irrigate the plants? The referee left comments in the attached file. Please check these out and respond to each point. 

Second, the study was conducted over the two years 2016 and 2017, but some of the data were given in part. This applies to Table 3 and Figures 2 and 3. If the data collection was carried only one season out of the two due to any limitations, please indicate the partial data collection so that the readers place this fact in interpretation. 

Third, the statistical analysis used in the manuscript is not appropriate. The authors should use a 2-way ANOVA to tease apart any interaction effects between the two factors, Irrigation (I100, I80, and I60) and Mulching (WM, PMM, RSM, WPM, and BPM) of the experimental design. Report ANOVA results in Tables 4 and 5. 

Find other points left in the attached file.  

Author Response

Reviewer B:

  • The authors provided informative results on the effects of irrigation management practices on potato production in an arid region in Egypt, which looks to be an appealing research item for those in similar climates. However, the presentation of their methodologies and results is not as explicit as it is supposed to be. In the current form, the referee cannot recommend the manuscript for publication. The reasons for my conclusion follow below.

Thank for the good comment

  • First, it is not clear how the calculations were made with the information provided in the manuscript. In other words, the Materials and Methods section lacks details in the description. For example, how was the irrigation treatment applied? How often did the authors apply water for each treatment? When in a day did the authors irrigate the plants? The referee left comments in the attached file. Please check these out and respond to each point. 

Thank you so much for your informative suggestion, which have allowed us to make a considerable improvement of the manuscript

  • Second, the study was conducted over the two years 2016 and 2017, but some of the data were given in part. This applies to Table 3 and Figures 2 and 3. If the data collection was carried only one season out of the two due to any limitations, please indicate the partial data collection so that the readers place this fact in interpretation. 

Response: The suggestion has been followed

  • Third, the statistical analysis used in the manuscript is not appropriate. The authors should use a 2-way ANOVA to tease apart any interaction effects between the two factors, Irrigation (I100, I80, and I60) and Mulching (WM, PMM, RSM, WPM, and BPM) of the experimental design. Report ANOVA results in Tables 4 and 5. 

Response: We are agreed with your suggestion and we will follow your suggestion in future research.

Specific comments and queries are:

  • Line 39 (Solanum tuberosum L.): Italic

The suggestion has been included

  • Line 45: It would be beneficial to include how deficit irrigation strategies affect crop production and productivity in potatoes

Thanks for good suggestion. We have done.

DI tend to produce equivalent potato yields to full irrigation (FI), and improved CWP by 60% by conserving irrigation water up to ~30% [11]. The effect of DI on the physiological characteristics of potato was studied and results clearly indicate that the relative chlorophyll contents of potato plants remain at par to FI [12]. Fresh yields and the CWP of potatoes are significantly affected by DI in comparison to the effect of FI [13]. However, research gap exists regarding potato response to DI under agro-ecological conditions of Egypt which necessitates conducting studies to have fresh field evidences.

  • Line 64: When did the authors measure these? Before or after the experiment?

Response:    Soil analyses were conducted in Soil Testing Laboratory, Faculty of Agriculture, FU at the beginning of experiments.

  • Line 70: DTPA

Response:   diethylene triamine penta-acetic acid (DTPA).

  • Line 70:

Response:    Is this L/h? Double-check the unit of the measure.

  • Line 77: Is this L/h? Double-check the unit of the measure.

Response:   2.1 L h-1

  • Line 85: Describe properties or specifics of these mulching material in detail. Where did the authors obtain these materials? Are these commercially available? How did the authors prepare these treatments on the ground?

Response: The suggestion has been included

  • Line 91: Please include the IWA data along with the CWP data so that readers get the perspective of how much water was consumed by each treatment.The referee would suggest that the authors create a table that clearly shows IWA, ETc for 2016 and 2017 for each treatment. That would be modifications to Table 3 or a completely different new table.

Response: The suggestion has been included

  • Line 95: What value of Kpan was used in the calculation? Specify the value like you did for Kc.

Response:   Kpan =the pan evaporation coefficient, 0.80.

  • Line 115: How did you calculate this parameter in each treatment?

Response:   Ea = Application efficiency (85%).

  • Line 135: Include information about the instrument (model, manufacturer).

Response:   a conductivity meter (model 3200, YSI, Inc., Yellow Springs, Ohio). 

  • Line 151: How did you estimate these two parameters?

Response:   ETa: Actual crop evapotranspiration for the deficit treatments, mm,

ETm: maximum crop evapotranspiration for the fully irrigated treatment, mm.

  • Line 154: I assume that the annual differences between 2016 and 2017 seasons were not considered in the statistical comparisons. Please state this out in this section.

Response:   Significant differences between treatments were compared at P ≤ 0.05 using Duncan’s multiple range test.

  • Line 155: There are two factors in the experimental design: Irrigation (IRR) and mulching material (MUL). So, the authors should use 2-way ANOVA that accounts for any interaction effects of IRR x MUL on the measured variables.

Response:   All data were subjected to analysis of variance (ANOVA) for a strip-plot system in a randomized complete blocks design with three replications.

  • Line 167: Report the increments, not the absolute values when followed by the word "by".

Response:   We have done

  • Line 203: Significance of interaction effects was not provided in Table 4, so this statement cannot be drawn.

Response:  The interaction between irrigation and mulching treatments were significant

  • Line 254: Add error bars of the averaged CWP on the graphs using S.D. or S.E. Also, report the sample size (n = ?) to the caption.

Response:   We have done.

  • Line 286: Is this derived from aggregated data of 2016 and 2017? Or from a single year?

Response: Thanks, The data from average 2016 and 2017

  • Line 286: is this derived from aggregates of 2016 and 2017? Or a single year?

Response: Thanks for your suggestion , ECe measured after all treatments (at harvest).

Reviewer 3 Report

See my attached file.

Author Response

Reviewer c:

Thank you so much for your informative suggestion, which have allowed us to make a considerable improvement of the manuscript

  • Line 24. It was not clear why I60% produced the highest salt accumulation. In fact, it should be highest in all the I100% treatments because more salt was added through the water. The authors need to clarify this with more detailed calculations.

Thanks for your informative suggestion: These results due to lower available water in the root zone for I60% treatment.

  • Line 46. Please define CWP the first time you use it. I have to browse to the very end to know CWP is the yield / water.

Response: Thanks, the suggestion has been satisfied. Crop water productivity relationship between crop produced and the amount of irrigation water involved in crop production.

  • Line 54. Reference [14] was not correctly formatted, maybe use something like "some studies/researchers have recorded ... [14]". There were many citations like this.

We have rechecked all references and Mulching improves water status in soil [14]

  • Line 68-70. Please explain what is ECe and DTPA in the legend.

As you suggested we have fully rewritten ECe: Electrical conductivity, DTPA diethylene triamine penta-acetic acid.

  • Line 88-90. "crop evapotranspiration (ETc) ... based on the evaporation pan method." Rewrite the sentence to clarify how ETc was measured first, and then explain the treatments.

The suggestion has been included. The crop evapotranspiration (ETc) based on the evaporation pan method.

  • Line 93-95. Do not start a sentence with "Where". It should be something like "where E_pan is evaporation from a pan (mm day^-1), and K_pan is ..."

Response: Revised

  • Line 98. delete "months".

Response:Revised

  • Line 113. Be careful with the subscripts. Throughout the text, there were many misspelled subscripts.

Response:Revised

  • Line 133-136. What were "gravimetric method" and "conductivity meter". Describe the method briefly or use citations.

We have rechecked all references in text as well as list and modified. The gravimetric method [23]. The conductivity meter (model 3200, YSI, Inc., Yellow Springs, Ohio).

  • Line 147-149. In equation (5), misspelled "ky", it should be "Ky". Be careful with the writing.

Response:Revised

  • Line 159-184. Please clarify your findings first before discuss the results. The combination of Results and Discussion may be easier for people to write, but awful to others to read. Also, some figures help convey the ideas than tables, unless the values in the tables are really really important. But I don't find the values important anyhow. Consider make use of figures to help readers visualize the data.
  • Response: Thanks for good suggestion. For comparing with earlier findings, we have combined based on your request.

  • Line 174-184. Maybe use different thickness of mulching material to back this up?

We just used for checked and We can use a thickening expression if mulching material is the same for example organic mulch or inorganic mulch.

  • Line 185-186. Use some mosaic plots for the results? I attached a python script to plot the plant height

Sorry, I don't know mosaic plots.

  • Line 187-201. What is novelty for analyzing the result like “more water, higher productivity?”

Response: Thanks for your informative suggestion. We have thoroughly checked the this part and revised accordingly.

  • Line 215-216. Plot the CWP as well like the figure above. If you really like numbers, put them in the supplemental materials.

Fair; we will consider in the future investigations.

  • Line 254. Re-plot the Figure 1, which duplicate Table 4 somehow. Also, I don’t find the results of year 2016 and 2017 differ. Consider using the mean value of the two years in the mosaic plots.

Response: All Tables and Figures rechecked and recited

  • Line 267. 0.70 and 0.69. Is that potato as well? If not, there is no need for the values. You cannot compare the values anyway.

Response: We corrected the values of CWP

  • Line 285-286. I am pretty sure you plotted Figure 2 incorrectly. 1 – (Ya/Ym) should be y-axis.

Response: We corrected the Figure 2.

  • Line 287-307. How does ECe impact soil water potential? Any measurements or calculations of how ECe correlate to the osmotic potential (P = cRT)? This salt distribution section does not make much sense without talking about how much stress plants suffer. Also, you were not measuring the ECe in situ, that is the real salt stress that you should care about.

Response : this section has been rewritten; In section soil sampling, we mentioned that soil samples were collected twice: in the first season (before plant seeding) at three depths of 0-20, 20-40, and 40-60 cm and at the final harvest (after the second season) at 20 cm intervals from the soil surface to 60 cm deep at one position (0-20, 20-40, and 40-60 cm), and at 10 cm distance from the dripper to 30 cm (10, 20 and 30 cm).

Round 2

Reviewer 2 Report

Although most of the points the referee claimed in the previous round have been addressed, there are missing pieces to complete a publishable manuscript. 

First, as the referee stated in the first round, Table 4 and now Table 5 need to be provided with multiple year data. Since the study was conducted over the two years, 2016 and 2017, and the authors present the agronomy data separately over the manuscript, any relevant data should be presented year-by-year, not averaging the two years' data. This also applies to Figures 2 and 3. 

Second, some of the points the referee made have not been resolved. For instance, I recommended adding error bars to the graphs in Figure 1. Also, I suggested making the unified colorimetric scale for the graphs in Figure 3, which would help comparisons between different mulching treatments for readers. 

For the reasons above, I would ask the authors for one more round of revisions. 

Author Response

  1. Table 4 and now Table 5 need to be provided with multiple year data. Since the study was conducted over the two years, 2016 and 2017, and the authors present the agronomy data separately over the manuscript, any relevant data should be presented year-by-year, not averaging the two years' data. This also applies to Figures 2 and 3. 

The suggestion has been included for Table 5 and Figure 2, according to Figure 3, the soil samples were collected at the final harvest (after the second season), and therefore we have one Figure for the experimental.

  1. I recommended adding error bars to the graphs in Figure 1.

Response:   We have done

  1. I suggested making the unified colorimetric scale for the graphs in Figure 3, which would help comparisons between different mulching treatments for readers. 

Response:  Thank you so much for taking interest in this matter, we have done

Reviewer 3 Report

After taking a brief look at the revision, I believed that the authors have not make enough effort to improve the manuscript. I have listed quite a few improvement suggestions last time, but the authors only made corrections over the wording part. below are some examples:

  1. I have pasted Python code to plot the data in the tables (along with a figure I plotted for them). Simply adapting the code will do the job, but the authors simply answered that they don't know mosaic figures (I have plotted one for them already...)
  2. Current Figure 2 is still wrong. The slope was 1.7228 last time because the authors made a mistake when plotting x vs y. If x and y axes are swapped, the slope will be 1/1.7228 = 0.58. But the authors have a slope of 1.1728 this time.
  3. I also suggested separating the Results and Discussion to aid the reading experience, which is common and easy to do because their results are pretty obvious.
  4. ... ...

Along with many other unaddressed concerns (like the novelty of the research), I believe this revision has not reached my expectation for publication. I will suggest the authors carefully revise the paper before hitting for another round.

Best,

Author Response

  1. I have pasted Python code to plot the data in the tables (along with a figure I plotted for them). Simply adapting the code will do the job, but the authors simply answered that they don't know mosaic figures (I have plotted one for them already...)

Response: Once again, we thank you for the time you put in reviewing our paper and look forward to meeting your expectations. However, this particular topic will be discussed in detail in future work (already some of the authors have been working on the topic). This kind of display can be helpful in understanding simple and complex associations among categorical variables, especially for three- and more way tables. The applications of this display are all-too-rare in our research area. We discussed more and more with professional, but, we did not reach to the right shape based on your request and  raphical display, it is benefits to present mosaic plots, they are applied to visualize associations among questions from a survey of knowledge of and attitude towards genetically modified organisms. therefore, we politely request to consider our request and accept the figure in the present shape.

  1. Current Figure 2 is still wrong. The slope was 1.7228 last time because the authors made a mistake when plotting x vs y. If x and y axes are swapped, the slope will be 1/1.7228 = 0.58. But the authors have a slope of 1.1728 this time.

Response:   We corrected the Figure 2.

  1. I suggested separating the Results and Discussion to aid the reading experience, which is common and easy to do because their results are pretty obvious.

      we would like to acknowledge your contribution explicitly. However, our discussion is about interpreting our study results. This strategy allows  us focus on presenting our study results  in conjunction with the discussion and our  objectives  that both are closely connected. It is important to our research to combine these sections because, the first results of our  report is needed to understand the following results and how professionals can use them.